# Epistatic Relationship between *MGV1* and *TRI6* in the Regulation of Biosynthetic Gene Clusters in *Fusarium graminearum*

**DOI:** 10.3390/jof9080816

**Published:** 2023-08-02

**Authors:** Kristina Shostak, Dianevys González-Peña Fundora, Christopher Blackman, Tom Witte, Amanda Sproule, David Overy, Anas Eranthodi, Nehal Thakor, Nora A. Foroud, Rajagopal Subramaniam

**Affiliations:** 1Ottawa Research and Development Centre, Agriculture and Agri-Food Canada, Ottawa, ON K1A 0C6, Canada; krishostak@gmail.com (K.S.); chris.blackman@agr.gc.ca (C.B.); tom.witte@agr.gc.ca (T.W.); amanda.sproule@agr.gc.ca (A.S.); david.overy@agr.gc.ca (D.O.); 2Lethbridge Research and Development Centre, Agriculture and Agri-Food Canada, Lethbridge, AB T1J 4B1, Canada; dianevys.gonzalezpenafundora@agr.gc.ca (D.G.-P.F.); anas.eranthodi@agr.gc.ca (A.E.); 3Department of Chemistry and Biochemistry, University of Lethbridge, Lethbridge, AB T1K 4M4, Canada; nthakor@uleth.ca; 4Department of Cell and System Biology, University of Toronto, Toronto, ON M5S 3B2, Canada; 5Department of Biology, University of Ottawa, Ottawa, ON K1N 6N5, Canada

**Keywords:** *Fusarium graminearum*, mitogen-activated protein kinase, secondary metabolites, RNA-seq

## Abstract

Genetic studies have shown that the MAP kinase MGV1 and the transcriptional regulator TRI6 regulate many of the same biosynthetic gene clusters (BGCs) in *Fusarium graminearum*. This study sought to investigate the relationship between *MGV1* and *TRI6* in the regulatory hierarchy. Transgenic *F. graminearum* strains constitutively expressing *MGV1* and *TRI6* were generated to address both independent and epistatic regulation of BGCs by *MGV1* and *TRI6*. We performed a comparative transcriptome analysis between axenic cultures grown in nutrient-rich and secondary metabolite-inducing conditions. The results indicated that BGCs regulated independently by *Mgv1* included genes of BGC52, whereas genes uniquely regulated by *TRI6* included the gene cluster (BGC49) that produces gramillin. To understand the epistatic relationship between *MGV1* and *TRI6*, CRISPR/Cas9 was used to insert a constitutive promoter to drive *TRI6* expression in the Δ*mgv1* strain. The results indicate that BGCs that produce deoxynivalenol and fusaoctaxin are co-regulated, with *TRI6* being partially regulated by *MGV1*. Overall, the findings from this study indicate that MGV1 provides an articulation point to differentially regulate various BGCs. Moreover, TRI6, embedded in one of the BGCs provides specificity to regulate the expression of the genes in the BGC.

## 1. Introduction

The regulatory hierarchy governing the expression of BGCs has been limited to a few genetic pathways in *F. graminearum* [1,2,3,4,5]. Studies have predominantly focused on the regulation of the BGC that produces the trichothecene mycotoxin deoxynivalenol (DON). The BGC23 harbors the majority of the genes involved in the production of DON, including the two regulatory genes *TRI6* and *TRI10* [6]. Targeted disruption of *TRI6* and *TRI10* significantly reduced the expression of genes within the cluster and abolished DON production [7]. Chromatin immunoprecipitation followed by Illumina sequencing (ChIP-seq) revealed that TRI6 is a global transcription factor that regulates more than 200 additional targets involved in carbohydrate metabolism and signal transduction [8]. Furthermore, TRI6 autoregulates its expression under nutrient-rich conditions and binds to promoter elements of some of the genes in the cluster under nutrient-limiting conditions. *TRI10* encodes a protein with no known functional domains, and unlike in *F. sporotrichioides*, *TRI6* expression levels were not significantly reduced in the Δ*TRI10* mutant in *F. graminearum* PH-1, although deletion of *TRI10* significantly reduced DON production [5].

Environmental and physiological cues, such as nutrient sources, light, and pH of the culture media, play an important role in the regulation of BGCs, and several common signal transducers and transcriptional regulators of many fungi have been characterized [9]. As an example, the target of the rapamycin (TOR) complex, conserved in all eukaryotes, integrates many of the nutrient cues and in two species of *Fusarium* (*F. fujikuroi* and *F. graminearum*); members of the TOR complex define the specificity to the expression of BGC genes [3,10]. The signals emanating from the TOR complexes are amplified by the mitogen-activated protein kinase modules (MAPK) [11]. The three MAPK pathways present in *Fusarium graminearum* have been ascribed distinct roles, with the HOG1 MAPK responsible for sensing and responding to the osmotic stress, while the MAP kinase module, MGV1, is linked to both monitoring and maintenance of the cell wall integrity (CWI) pathway [12,13]. An orthologue of MGV1, GPMK1 represents the third MAPK pathway that is responsible for filamentous growth and sexual reproduction [14,15]. All three of these pathways are important for vegetative growth, reproduction, and pathogenicity in *F. graminearum*; however, evidence suggests that only HOG1 and MGV1 are major contributors to the regulation of BGC in *F. graminearum* [12]. Although targeted deletion of *HOG1* and *MGV1* shows reduced virulence, a phenotype related to reduced fitness, the impact on the production of secondary metabolites is distinct [12]. Both exhibit contrasting phenotypes in the production of DON; genetic studies show that MGV1 positively impacts DON production, while HOG1 has the opposite effect. Similar contrasting phenotypes have been observed with aurofusarin and zearalenone biosynthesis [12]. It is not currently known how the distinct regulations are achieved, except to suggest that many of the targets of the various MAPKs are regulated post-translationally and, thus, complex regulatory mechanisms are in play.

The signals from the MAPKs and other signaling modules are parsed through many transcriptional regulators that act either independently, synergistically, or epistatically to the various signaling modules [16,17]. One of the well-characterized families of transcription regulators, AreA, plays a pivotal role in integrating nitrogen metabolism [18]. AreA is responsive to nitrogen sources; regulated by the cyclic adenosine monophosphate (cAMP)—protein kinase A (PKA) pathway, it has been shown to interact with TRI10 and regulate the DON pathway in *F. graminearum* [5]. The conspicuous absence of regulation in zearalenone production by AreA underscores the distinctiveness of the regulation of BGCs by the various signaling modules. Additional support for this distinctiveness comes from studies of other transcriptional regulators, such as the velvet complex proteins that perceive light signals and PacC, which perceive changes in pH in the culture medium to regulate an array of BGCs, including the biosynthesis of DON [19,20].

To distinguish the regulation of BGCs by the MGV1 signaling pathway and the transcriptional regulator TRI6 in *F. graminearum*, we undertook comparative transcriptomic studies with deletion and constitutive expression of *MGV1* and *TRI6*. Our results show that *MGV1* broadly regulates many BGCs, while *TRI6* distinguishes the BGC’s regulation towards the production of various metabolites.

## 2. Materials and Methods

### 2.1. F. graminearum Strains and Culture Conditions

*F. graminearum* (NRRL29169) was the wild-type (WT) and parental strain used for the construction of the transgenic strains. The construction of *∆mgv1* and *MGV1*Oex6 strains have been previously described [21,22]. Briefly, overexpression was achieved by replacing the native promoter of *MGV1* with the constitutive promoter (p*GPD*) of the *glyceraldehyde 3-phosphate dehydrogenase* gene from *A. nidulans* [22]. The construction of other MGV1Oex lines was constructed similar to *MGV1*Oex6 [22,23]. The macroconidia were produced in carboxymethylcellulose media and used as the inoculum for all cultures at a concentration of 5 × 10^3^ spores mL^−1^. Mycelia were grown in either preferred nutrient medium (PN) (56 mM NH_4_Cl, 8.1 mM MgSO_4_ 7H_2_O, 0.23 mM FeSO_4_·7H_2_O, 14.7 mM KH_2_PO_4_, 2 g L^−1^ peptone, 2 g L^−1^ yeast extract, 2 g L^−1^ malt extract, and 111 mM glucose) or in nonpreferred nutrient (NPN) medium (6.2 mM putrescine dihydrochloride, 22 mM KH_2_PO_4_, 0.8 mM MgSO_4_·7H_2_O, 85.6 mM NaCl, 116.8 mM sucrose, 108.6 mM glycerol, pH 4.0). Unless otherwise specified, all strains in the study were cultured in the dark, at 28 °C, with shaking at 160 rpm for 24 h. For nonpreferred nutrient (NPN) conditions, strains were first cultured in PN for 24 h, as described, followed by washing and growth in a DON-producing medium, as previously described [24].

### 2.2. Construction of the Δmgv1/TRI6OX Strain by CRISPR-Cas9

Homologous-based recombination (HDR) using CRISPR/Cas9 in *F. graminearum* was based on previous studies [25,26]. Briefly, microhomology primers flanking *MGV1* were used for the construction of a hygromycin repair template to replace the *MGV1* gene (Figure 1A and Appendix A). Another set of microhomology primers flanking a 225 bp portion of the TRI6 promoter was used to construct the repair template containing 1016 bp of GAPDH promoter to functionally replace the native *TRI6* promoter (Figure 1B and Appendix A). The upstream and downstream gRNAs associated with each genomic modification and 4 µg of each of the repair templates were incubated with 100 µL of protoplasts. Protoplast preparation, regeneration, and transformant selection were performed as before [24,25]. Deletion of the *mgv1* gene and the *TRI6* promoter replacement was confirmed using whole genome sequencing (Appendix A). The transformants were sequenced with Illumina Novaseq Shotgun DNA sequencing (https://www.genomequebec.com, accessed on 12 September 2021). Sequencing reads were aligned to the reference genome of NRRL29169 in CLC-Genomics Workbench (v20) using the default settings and examined for large InDels and the absence of the gene of interest [25]. Gene expression was confirmed using reverse transcriptase quantitative PCR (RT-qPCR) under PN conditions (Appendix A).

### 2.3. Gene Expression Analysis by RNA Sequencing

*F. graminearum* wild-type, *∆mgv1*, *MGV1*Oex1, *MGV1*Oex6*,* and *∆mgv1*::*TRI6*OX strains were grown in 4 mL of liquid culture under NPN conditions as described [24]. Total RNA from three biological replicates per strain was extracted using Trizol reagent, as described previously [8,25]. Total RNA was sequenced using Illumina NovaSeq 6000 platform (https://www.genomequebec.com accessed on 12 September 2021). The raw reads were trimmed and aligned to the *F. graminearum* gene coding sequences predicted in NRRL29169 assembly (GWAS Accession# SPRZ00000000) using the RNA-Seq Analysis feature in CLC Genomics Workbench, version 12, with the following parameters: mismatch cost 2, indel cost 3, length fraction 0.9, similarity fraction 0.8, the maximum number of hits for a read 10. Unique gene read counts were manually curated for genes with a minimum of 50 unique gene reads. Filtered counts were then imported into the R environment and normalized using default parameters in DESeq2, which uses negative binomial models to represent the number of reads assigned to a gene in a sample. Differential expression analysis was also carried out using DESeq2, with the threshold fold change ≥2 and the FDR-adjusted *p*-value ≤ 0.05. The normalized reads and differential expression genes (DEGs) are included in the Appendix A.

### 2.4. Quantitative PCR (RT-qPCR) Analyses

The gene expression from the RNA-sequencing (RNA-seq) data was confirmed using RT-qPCR. One microgram of total RNA was reverse-transcribed using a high-capacity cDNA reverse transcription kit with random hexamer primers (Applied Biosystems, Waltham, MA, USA). The qPCR was performed using the Applied Biosystems PowerUp SYBR Green reaction mix and QuantStudio 3 Real-Time PCR system (Thermo Fisher, Nepean, ON, Canada). The relative expression was calculated using the Pfaffl method with *EF1α* (FGSG_08811) and *β-tubulin* (FGSG_09530) as reference genes [27]. Significance was determined using the Student’s *t*-test at *p* ≤ 0.05. The primers used for the amplification are listed in Appendix A.

### 2.5. Mycelial Growth and Reproductive Structures Assays

Vegetative growth, spore germination, and perithecia formation were measured for each of the *MGV1*Oex transformants and the WT. For vegetative growth, the fungus was grown on potato dextrose agar (PDA) plates with a mycelial plug as the inoculum. The growth diameter was measured every 24 h until the mycelia reached the edge of the plate. Eight plates were evaluated for each strain, and the experiment was repeated twice (n = 16/experiment).

Macroconidia spore germination was measured on synthetic nutrient agar (Spezieller Nährstoffarmer Agar; SNA) at 27 °C. Each plate was marked with 0.7 × 0.7 cm^2^, and 8 squares (replicates) were inoculated with 3 μL of 10^4^ macroconidia mL^−1^. Germination was observed in each square using a light microscope (Leica DM500, Concord, ON, Canada) at 100× magnification. At 3, 6, and 8 h, the number of germinated spores (when the germ tube exceeded the size of the conidia) was counted under a light microscope (Leica DM500) at 100× magnification. The experiment was performed in triplicate (n = 24). Data were reported as the percent macroconidia germination and analyzed with one-way ANOVA (analysis of variance), and the means were compared using the Tukey test, *p* ≤ 0.05.

Perithecia formation was determined on carrot agar inoculated with mycelial plugs, and after 4 d of incubation at 28 °C, the mycelia were removed. Perithecia formation was induced with the application of 1.0 mL (*v*/*v*) of 2.5% Tween 60 to the agar surface, and the plates were incubated at 25 °C with continuous light; this step was repeated if after 48 h the mycelia reappeared. The presence of perithecia was determined after 3–5 d by disrupting the perithecium with a cover slide to release the ascospores, which were then observed under a light microscope (Leica DM500) at 100× magnification. The experiment was conducted in triplicate.

### 2.6. Quantification of 15-Acetylated DON (15-ADON) Accumulation in Axenic Cultures

The 15-ADON accumulation analyses in Δ*mgv1*, *MGV1*Oex strains, and WT were performed using a two-stage liquid media protocol [24]. A suspension of 10 μL containing 2 × 10^4^ spores was inoculated into a Falcon Multiwell 6-well culture tray containing 4 mL of first-stage growth media at pH 7.0. Each well contained an autoclaved nylon net filter (100 μm NY1H type, Millipore, Burlington, MA, USA). The trays were sealed with parafilm and affixed to an orbital shaker, for 24 h in the dark, at 170 rpm. After 24 h, the liquid was removed using a sterile transfer pipette without disturbing the filter with the mycelial growth, and it was resuspended in sterile water to remove traces of first-stage media. This step was repeated, and the mycelia/filter was then resuspended in 4 mL of second-stage media, pH 4.0 [8]. The mycelium was grown in the second stage under the same conditions described above, and the supernatant was collected after 48 h.

Mycelial solids were collected and dried under a vacuum for 24 h before weighing. A nylon filter was used as a blank measurement. The sample supernatant was first filtered (0.2 μm) and combined 450 μL sample + 150 μL MeOH in the HPLC vial. Trichothecenes were analyzed on a Shimadzu prominence LC-20AD (Mandel) with 100 μL injection on a Shimadzu SIL-20A HT prominence autosampler. The samples were run on a Restek Pinnacle DB C18 Column (5 um, 150 × 4.6 mm) using a 22.5% isocratic MeOH: H_2_O flow over 20 min at a rate of 1 mL min^−1^. Trichothecenes were monitored by UV 220 nm [8,23].

### 2.7. Immunodetection of MAPKs

Protein was extracted from the mycelia of Δ*mgv1*, *MGV1*Oex, and WT strains collected from four-day cultures on PDA of or from liquid culture under NPN conditions as described above. For each strain and condition, samples were collected in triplicate for one protein extraction, and the experiment was repeated three times. Mycelia were ground to a powder in liquid nitrogen using a mortar and pestle and transferred to a tube with 10 mL MAPK extraction buffer per gram of mycelia. MAPK extraction buffer was composed of 1 M NaCl, 50 mM NaF, 1 mM phenylmethylsulfonyl fluoride (PMSF), 1 mM Na_3_VO_4_, protease inhibitor cocktail (Thermofisher, Halt™ Protease Inhibitor Cocktail, 100 µL per 10 mL buffer), 0.2% β-mercaptoethanol (BME), and 50 mM sodium phosphate buffer, pH 8. The samples were incubated at 4 °C with gentle agitation for 1 h. The cell debris was removed by centrifugation in 1.5 mL tubes at 13,751× *g* for 5 min at 4 °C. The supernatant was filtered through a 0.45 µm syringe filter. The protein concentration was measured using the Bradford reagent (BioRad, USA).

Ten micrograms of total protein were denatured with 1 × NuPAGE^TM^ LDS Sample Buffer (ThermoFisher) and 0.5% (*v*/*v*) β-mercaptoethanol at 70 °C for 10 min. The samples were resolved on 12-well Bolt^TM^ 4–12% Bis-Tris-Plus gels (Invitrogen) in Bolt^TM^ MOPS SDS Running Buffer (Invitrogen) at 150 V for 1.5 h. The gels were run in duplicate, where one was stained with OmniPur Coomassie Blue R-250 (EMD Millipore) to visualize the resolved proteins, and the other was used to transfer proteins onto Immun-Blot^®^ PVDF Membranes (BioRad, Saint-Laurent, QC, Canada). Protein transfer was carried out by electrophoresis in 25 mM Tris, 192 mM glycine, pH-8.3, and 20% methanol at 100 V for 1.5 h at room temperature. The uniform transfer was verified with the Pierce^TM^ Reversible Protein Stain Kit for PVDF Membranes, according to the manufacturer’s instructions. After stain removal, membranes were blocked with 3% bovine serum albumin in Tris-buffered saline (TBS) for 1 h and then probed overnight at 4 °C with 10 mL of 1:1000 dilutions of primary antibody in 5% BSA in TBST (1x TBS buffer with 0.05% Tween20). Four different primary polyclonal antibodies were used to detect different MAPKs: (1) anti-phospho-p44/42 MAPK (ERK1/2) (Cell Signaling Technology, Danvers, MA. USA) detects phosphorylated forms of the *Fusarium* MAPKs MGV1 (p-MGV1) and GPMK1 (p-GPMK1); (2) anti-MAP kinase (ERK1/2) (Sigma) detects total MGV1 and GPMK1; (3) anti-phospho-P38 MAPK (Cell Signaling Technology, USA) detects phosphorylated forms of the *Fusarium* p38 MAPK, p-HOG1; (4) anti-P38 MAPK (Cell Signal, USA) detects unphosphorylated forms of HOG1. Anti-α tubulin YOL 1/34 (Santa Cruz Biotechnology) from rats and anti-actin (Abcam, Waltham, MA, USA) from mice were used to detect the *Fusarium* housekeeping protein tubulin and actin, respectively. Following three washes in TBST, the membrane was incubated for 1 h at room temperature with 12 mL of Goat anti-rabbit-HRP, Goat anti-rat-HRP, or Goat anti-mouse-HRP conjugated secondary antibodies (BioRad), and 1:10,000 Precision Protein^TM^ StrepTactin-HRP conjugate (BioRad) in 3% BSA-TBST. The blots were developed with SuperSignal^TM^ West Pico PLUS Chemiluminescent Substrate.

### 2.8. FHB Disease Assays

FHB assays were carried out in Canadian wheat cultivars, ‘Roblin’ (highly susceptible) and ‘Penhold’ (moderately resistant). Plants were grown at 22 °C/18 °C (day/night) in a greenhouse with a 16 h photoperiod. At anthesis, one spike per plant was point inoculated by pipetting 10 µL of inoculum (1 × 10^5^ macroconidia mL^−1^ in 0.2% Tween20) into a single floret. After inoculation, plants were incubated in a 25 °C (day/night) misting growth cabinet (95% relative humidity) with a 16 h photoperiod for 3 days and then returned to their original growing environments. Disease rating was performed at 7-, 12-, and 18-days post inoculation (dpi) by counting the number of diseased spikelets below and including the inoculated spikelet. The experiment was repeated three times over four weeks. The mean values of five plants per inoculum were calculated for each repetition using one-way ANOVA followed by Tukey’s HSD test (α = 0.05).

## 3. Results

The NPN (nonpreferred nutrient) conditions have previously been documented to activate a plethora of BGCs, including BGC23, which is responsible for DON production (24). An outline of the scheme is presented to navigate the various RNA-seq datasets that were used to distinguish the expression of BGCs influenced by *TRI6* and *MGV1* under NPN conditions (Figure 2). The NPN acts as a stimulus to trigger the MAPK, MGV1, which can activate BGCs (Pathways A and B, Figure 2) or indirectly through the action of *TRI6* (Pathway B, Figure 2). The stimulus can also activate BGCs that are independent of *MGV1*, but dependent on *TRI6* (Pathway C, Figure 2). Finally, the BGCs can be co-regulated by both MGV1 and TRI6 (Pathway D, Figure 2)

To obtain the genes specifically regulated by *MGV1*, we constructed four *MGV1*Oex strains (Oex1, Oex2; Oex6, and Oex 8), and under noninducing (or PN) conditions, all four strains exhibited constitutive expression and phosphorylation of MGV1 compared to the WT strain (Figure 3A,B). In response to stimulus (NPN condition), MGV1 is activated in both the WT and *MGV1*Oex6, and these conditions do not alter the expression of the other two *F. graminearum* MAPKs, Gpmk1 and Hog1 (Figure 3C). The four strains with active MGV1 exhibited similar growth properties as the WT (mycelia and germination; Appendix A). In contrast, all four strains produced higher levels of 15-ADON in the culture medium (Appendix A). Interestingly, the increased levels of 15-ADON in the four strains did not translate into an increase in disease spread in either a susceptible wheat variety (cv. Roblin) or an intermediate resistance wheat variety (cv. Penhold) (Appendix A).

We performed RNA-seq analysis with WT, Δ*mgv1*, and the two *MGV1*Oex strains (Oex1 and Oex6) grown in NPN conditions. Normalized reads and DEGs are shown in Appendix A. To obtain *MGV1*-specific regulated genes, we developed a common dataset of differentially regulated genes (DEGs) of the two *MGV1*Oex strains and the Δ*mgv1* strain (Figure 4A). The overlap dataset with 1120 DEGs represented *MGV1*-regulated genes of the two Δ*mgv1*-*MGV1*Oex strains (1207 in *MGV1*Oex1 and 1185 in *MGV1*Oex6) (Figure 4B, Appendix A). A FunCat analysis of 1120 DEGs encompassed genes from both primary and secondary metabolisms, cellular transport, and genes involved in virulence (Appendix A). The RNA-seq was validated by RT-qPCR experiments with six genes that were either positively or negatively regulated by *MGV1* (Figure 5).

The 1120 DEGs comprised 107 BGC genes (Figure 6A; Appendix A). These included positive regulation of BGCs 23, 49, and 64 by *MGV1*, involved in the production of 15-ADON, butanolide, and fusaoctaxin, respectively, while aurofusarin (BGC13) and fusarin (BGC42) and BGCs 06, 09, 12, 37, 51, 52, and 62 with no known identified products were negatively regulated by *MGV1* (Appendix A). It is noteworthy to indicate that there was a 5-fold downregulation of *TRI6* and concomitant reduction in *TRI5* expression in the Δ*mgv1* mutant strain, suggesting that *MGV1* partially regulates the expression of *TRI6*, which will impact the expression of BGC23 and other BGCs regulated by *TRI6* (Figure 5) [24].

Similarly, to obtain the genes specifically regulated by *TRI6*, a dataset of DEGs from the RNA-seq analysis between Δ*tri6* and WT grown in NPN conditions was compared to the dataset of DEGs from RNA-seq analysis between *TRI6*OX and the Δ*Tri6* strain grown under the same conditions. This dataset has been published and the analysis of the 687 DEGs will only be briefly summarized to suit this study (Figure 6A, Appendix A) [24]. We observed 110 BGCs that are differentially expressed (Figure 6A; Appendix A). Similar to *MGV1*, *TRI6* positively regulates BGCs that produce 15-ADON, butanolide, and fusaoctaxin and negatively influences the production of gramillin, aurofusarin, and fusarin (Appendix A).

To obtain genes co-regulated by *MGV1* and *TRI6*, we compared the 1120 DEGs regulated by *MGV1* and 687 DEGs regulated by *TRI6* (Figure 6, Appendix A). The overlap of two datasets represented 228 co-regulated genes, representing 51 BGCs that included the positively regulated trichothecene gene cluster (BGC23), butanolide gene cluster (BGC49), and fusaoctaxin (BGC64), the negatively regulated aurofusarin gene cluster (BGC13). In addition to genes that are co-regulated, we also identified 892 DEGs with 56 BGCs that are *MGV1* dependent but *TRI6* independent (Appendix A). This category contained only ‘orphan’ genes of a BGC; we defined ‘orphan’ genes as those that are part of the BGC but have different co-expression patterns. For example, only three (FGSG_03529, FGSG_03531, FGSG_03533) of the ~15 genes of the trichothecene gene cluster (BGC23) are represented in this category and their biochemical function in the cluster have not been properly elucidated. These ‘orphan’ genes are likely regulated independently of the MGV1 pathway. The Venn diagram also showed 459 DEGs comprising 59 BGCs that are regulated independently of *MGV1* but are *TRI6* dependent (Appendix A). The majority of genes included in this category are represented in BGC02, (responsible for gramillin production), BGC16, and BGC66 (chrysogine production).

Since epistasis between MGV1 and TRI6 was alluded to, we were interested to differentiate the regulation of BGCs by the two genes. Therefore, we constitutively overexpressed *TRI6 in locus* in the Δ*mgv1* mutant strain background (Δ*mgv1*::*TRI6*Oex). The strain was constructed by simultaneous insertion of the hygromycin resistance marker in the *MGV1* locus and a ~1 kb promoter of glyceraldehyde 3-phosphate dehydrogenase (p*GPD*), a constitutive promoter, upstream of the *TRI6* start site. CRISPR/Cas9 was used to make the double insertion construct, which was verified by both PCR and whole genome sequencing (Appendix A); we used RT-qPCR to confirm that the expression *TRI6* was constitutive (Appendix A). We characterized this strain with respect to growth properties in three different media and, as observed, the constitutive expression of *TRI6* did not significantly alter the phenotype observed with that from the Δ*mgv1* strain (Appendix A). This demonstrates that the fitness loss linked to *MGV1* disruption cannot be rescued by overexpression of *TRI6* and, therefore, is not linked to *TRI6* [13].

An RNA-seq experiment was performed with the Δ*mgv1*::*TRI6*Oex strain in the NPN condition and was compared to the WT strain grown in the same conditions. A total of 969 DEGs represented genes regulated by *TRI6* that were independent of *MGV1* (Appendix A). Among these DEGs, 108 represented genes of the BGCs involved in the production of gramillin (BGC02), aurofusarin (BGC13), orcinol (BGC18), fusarin (BGC42), fusaristatin (BGC47), butanolide (BGC49), and chrysogine (BGC66) (Appendix A).

To establish epistasis, the dataset was compared to the 1120 DEGs that represented *MGV1*-specific regulated genes (Figure 4). The Venn diagram (Figure 7) differentiated those DEGs that are regulated by *MGV1* (Pathways A and B) and those that are regulated by *TRI6* (Pathways C and D) (Figure 2). The 24 BGCs regulated by *MGV1* alone represent the ‘orphan’ genes of a BGC. In contrast, the 84 BGCs regulated by *TRI6* (independently or through *MGV1*) represent a full complement of genes of a given BGC, including those involved in the production of gramillin, aurofusarin, fusarin, fusaristatin, butanolide, etc. (Appendix A).

## 4. Discussion

MAPKs are essential for the transduction and amplification of environmental signals to the transcriptional regulatory network. Disruption mutants of the MAPK, *MGV1*, indicate that this gene is involved in the regulation of DON biosynthesis in *F. graminearum* [21]. The transcription factor, TRI6, regulates the expression of genes of the trichothecene biosynthesis genes (*TRI* genes) [9]. A connection between MGV1 and TRI6 in this regulation has not previously been investigated. Notably, both genes have also been implicated in other metabolic pathways and BGCs [21,24]. Here, we explored the interaction between *MGV1* and *TRI6* in the regulation of BGCs in *F. graminearum*. Comparative transcriptomics was used with various genetic mutants of *MGV1* and *TRI6* to reveal BGCs that they are independently regulated by the two genes and those BGCs that are dependent on both regulators. Our previous study [24] that combined transcriptome and metabolome suggested that *TRI6*-regulated BGCs were involved in the production of 15-ADON, fusaoctaxin A (BGC64), and gramillin (BGC02). The regulatory hierarchy governing the biosynthesis of secondary metabolites constructed with the RNA-seq datasets was able to differentiate BGCs controlled by *MGV1* and *TRI6*. The present study was able to discern the regulation of the aforementioned BGCs with BGCs 23 and 64 being co-regulated, while BGC02 was regulated by *TRI6*, independent of *MGV1* (Figure 8, Appendix A).

As related in the hierarchy scheme (Figure 2), a combination of A and B, comprising of the *MGV1*-activated genes, include BGC18 involved in the production of Orcinol and other uncharacterized BGCs. The role of the products synthesized by these uncharacterized BGCs in plant pathogenesis or against other microbes is yet to be determined, but CRISPR technology is facilitating our endeavor in an accelerated fashion. The *TRI6* independent pathway (pathway C) includes BGCs that produce gramillin (BGC02), fusaristatin (BGC47), and an uncharacterized BGC16 (Figure 8, Appendix A). The BGCs that are epistatic include clusters that produce aurofusarin (BGC13), triacetylfusarinine (BGC21), trichothecene (BGC23), butenolide (BGC49), chrysogine (BGC66), an unknown product synthesized by an NPS (Non-ribosomal peptide synthetase) belonging to BGC72 (Appendix A; Pathway D, Figure 2). Since the expression of *TRI6* is downregulated by 5–6 fold in the Δ*mgv1* mutant strain (Figure 5), it is more than likely that the epistasis is through the regulation of *TRI6* by *MGV1*. The study also underpins the importance of TRI6 as a global regulator by influencing other BGCs [8]. The next step would be to understand the mechanism by which TRI6 regulates these BGCs. Our previous study using chromatin immunoprecipitation demonstrated that under PN and NPN conditions, TRI6 is only bound to the promoters of the BGC23 and not to other co-regulated clusters such as BGC47 or BGC02 [24]. This suggested that TRI6 imposes its regulation of other BGCs indirectly through its interactions with other regulators such as GRA2 [24].

It is unclear how the regulation of TRI6 is achieved; post-translational modification through phosphorylation of TRI6, is unlikely as phosphoproteome studies in *F. graminearum* did not reveal any such modifications and no predictive MAPK phosphorylation domains are present in the translated sequence [28]. TRI6 has been proposed to be regulated partially by TRI10, another regulator in the BGC23, involved in the production of 15-ADON, however, a recent study demonstrated that this may be due to the structural changes of BGC23 and not necessarily through the action of *TRI10* [29].

Overall, the analysis of the BGCs reveals that many BGCs are regulated through MGV1, and a subset is finely tuned by the regulator TRI6 embedded within a BGC. It is more than likely that other BGCs in the A/B pathways are similarly fine-tuned by the other transcriptional regulators embedded with that respective BGC. As noted, the three MAPKs in *F. graminearum* have overlapping functions with respect to the regulation of 15-ADON production (BGC23) [14]. The observations reinforce the idea that MAPKs including MGV1 serve as an articulation point to various pathway-specific regulators. The molecular mechanism, either through differential phosphorylation of transcriptional regulators or through protein interactions, will be the next step to understanding how MGV1 differentiates various signaling pathways.

We also observed that both MGV1 and TRI6 act as positive and negative regulators. For example, TRI6 positively regulated genes in the BGC16, while negatively regulating genes of the BGC that produce gramillin (BGC02), and fusaristatin (BGC47) (Appendix A). This may be reflective of the reallocation of resources within the fungus but may also represent antagonistic interactions that may be required to acquire full pathogenicity potential. Efforts are underway to delete the representative core gene in each BGC and characterize their virulence function individually and in combination.

In conclusion, our efforts to study the epistatic relationship between MGV1 and TRI6 were achieved through the manipulation of genes by the CRISPR/Cas9 approach. A traditional approach through sequential deletion/insertion would have been problematic as the Δ*mgv1* mutant strain with its fitness defect was recalcitrant to protoplasting (weak cell wall) and subsequent regeneration steps. Thus, the CRISPR technique allowed us to achieve the dual manipulation of *MGV1* and *TRI6* expression using WT protoplasts, sidestepping the fitness defect of the Δ*mgv1* mutant strain.

## Figures and Tables

**Figure 1 jof-09-00816-f001:**
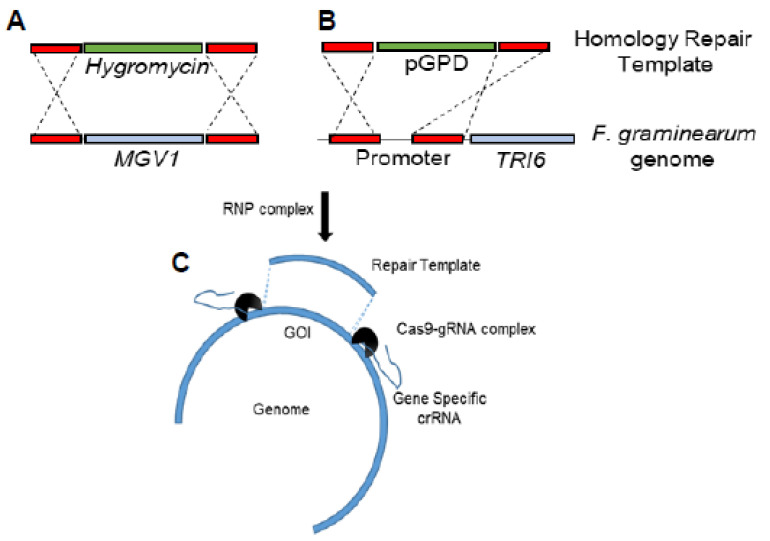
A scheme to construct *F. graminearum* Δmgv1/TRI6OX strain. Two repair templates were constructed: (**A**) first repair template consisted of the selection marker hygromycin that will replace the coding region of MGV1; (**B**) second repair template consisted of a ~1 kb promoter of GAPDH (pGPD) that will replace the TRI6 promoter; (**C**) *Fusarium* protoplasts were incubated with the two repair templates with the ribonucleoprotein (RNP) complex to promote a homologous recombination event.

**Figure 2 jof-09-00816-f002:**
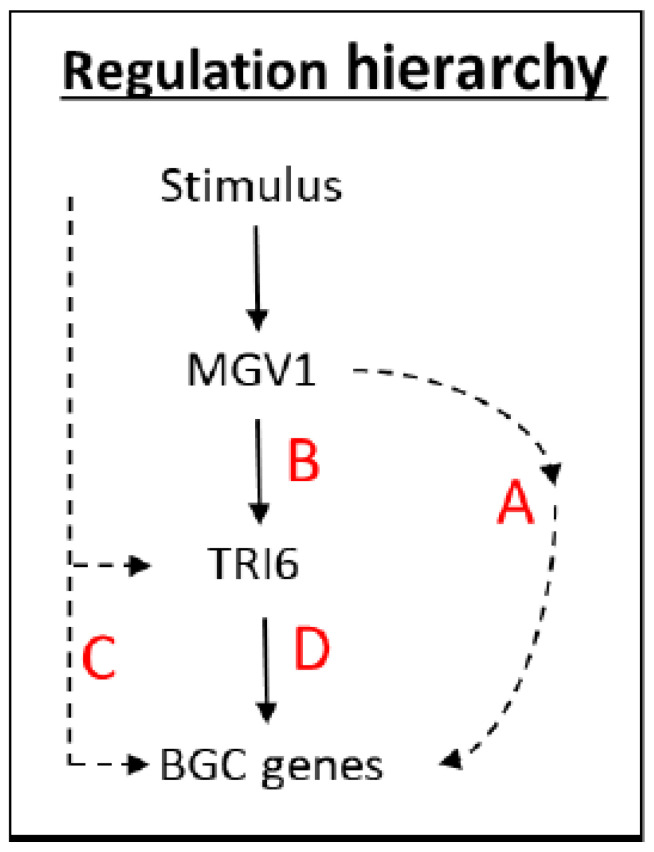
Schematic representation of hierarchical regulation of BGCs by MGV1 and TRI6. The NPN acts as a stimulus to trigger the MAPK, MGV1, which can activate BGCs (Pathways A and B) or indirectly through the action of TRI6 (Pathway B, Figure 1). Similarly, the stimulus can also regulate TRI6 function through a pathway that is independent of MGV1 (Pathway C, Figure 1) or dependent on MGV (Pathway D, Figure 1).

**Figure 3 jof-09-00816-f003:**
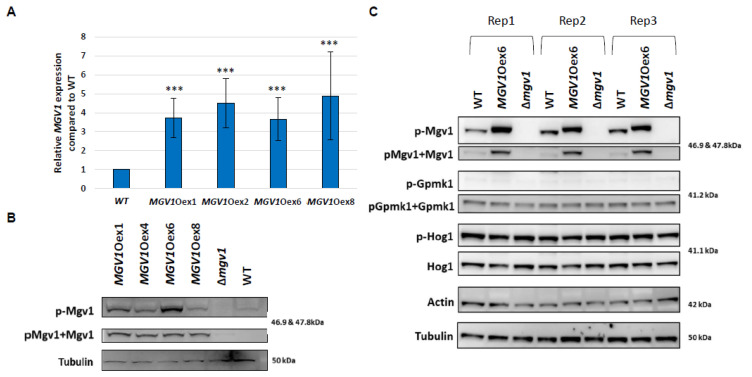
Differential expression and activation of MAPKs in WT, Δmgv1, and MGV1Oex strains. (**A**) RT-qPCR confirmation of increased Mgv1 expression in MGV1Oex strains. Relative expression was calculated using the Pfaffl method with EF1α (FGSG_08811) and β-tubulin (FGSG_09530) as reference genes. Bars represent a mean of three biological replicates with error bars representing standard error. Significance was determined by Student’s *t*-test at *p* ≤ 0.05; *** *p* < 0.001 (**B**) Activation of the Mgv1 enzyme in the MGV1Oex strains. Immunodetection of phosphorylated Mgv1 (pMgv1) and total Mgv1 (pMgv1 and unphosphorylated forms) and the housekeeping protein tubulin from mycelium-grown PDB. One of three replicates was presented. (**C**) Activation of MAPKs in different strains cultured in NPN. Immunodetection of phosphorylated Mgv1 (pMgv1) and total Mgv1 (pMgv1 and unphosphorylated forms), phosphorylated Gpmk1 (pGpmk1) and total Gpmk1 (pGpmnk1 and unphosphorylated forms), phosphorylated Hog1 (p-Hog1), unphosphorylated Hog1, and the housekeeping protein tubulin from mycelium grown in PNP. Triplicate results are presented in one blot. Note that the Mgv1 protein in Mgv1Oex strains carries a His-tag, which increased the molecular weight of Mgv1 from 46.9 kDa to 47.9 kDa in the overexpression strains.

**Figure 4 jof-09-00816-f004:**
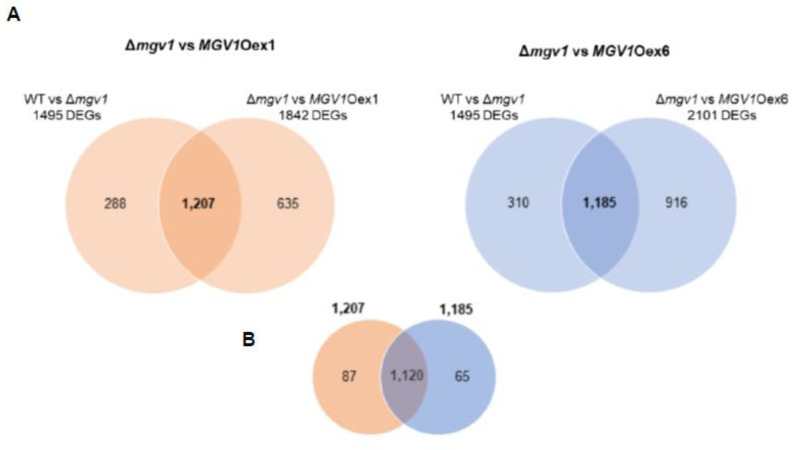
Venn diagram showing differentially expressed genes (DEGs) to identify MGV1-specific regulated genes. Venn diagram showing differentially expressed genes (DEGs) to identify MGV1-specific regulated genes. (**A**) DEGs from wild-type and Δmgv1 strains grown in NPN conditions (1495) were compared to DEGs of the Δmgv1 and the two in-locus *MGV1* overexpression strains (Oex1; 1842; and Oex6; 2101) strains grown in NPN conditions. (**B**) An overlap (1120 DEGs) that are common to both overexpression strains, representing MGV1 regulated genes.

**Figure 5 jof-09-00816-f005:**
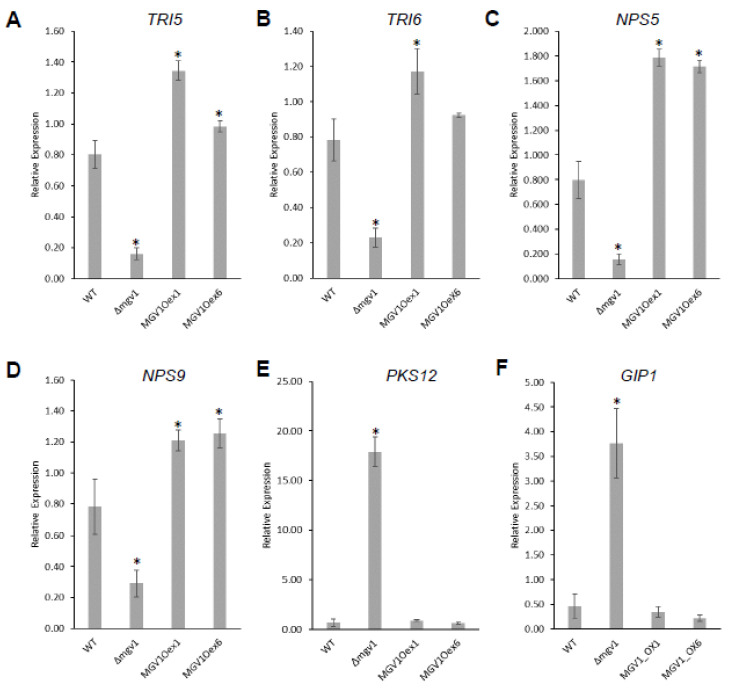
MGV1 affects the expression of BGC genes. The wild-type (WT), ∆*mgv1*, *MGV1*Oex1, and *MGV1*Oex6 strains were grown in NPN conditions for 24 h, total RNA was isolated, and RT-qPCR was performed with six genes that are differentially expressed in the RNA-seq analysis. TRI5, TRI6, NPS5, and NPS9 are positively regulated by MGV1 (**A**–**D**), PKS12 and GIP1 are negatively regulated (**E**,**F**) Relative expression was calculated using the Pfaffl method with EF1α (FGSG_08811) and β-tubulin (FGSG_09530) as reference genes. Bars represent a mean of three biological replicates with error bars representing standard deviation. Significance (*) was determined with Student’s *t*-test at *p* ≤ 0.05.

**Figure 6 jof-09-00816-f006:**
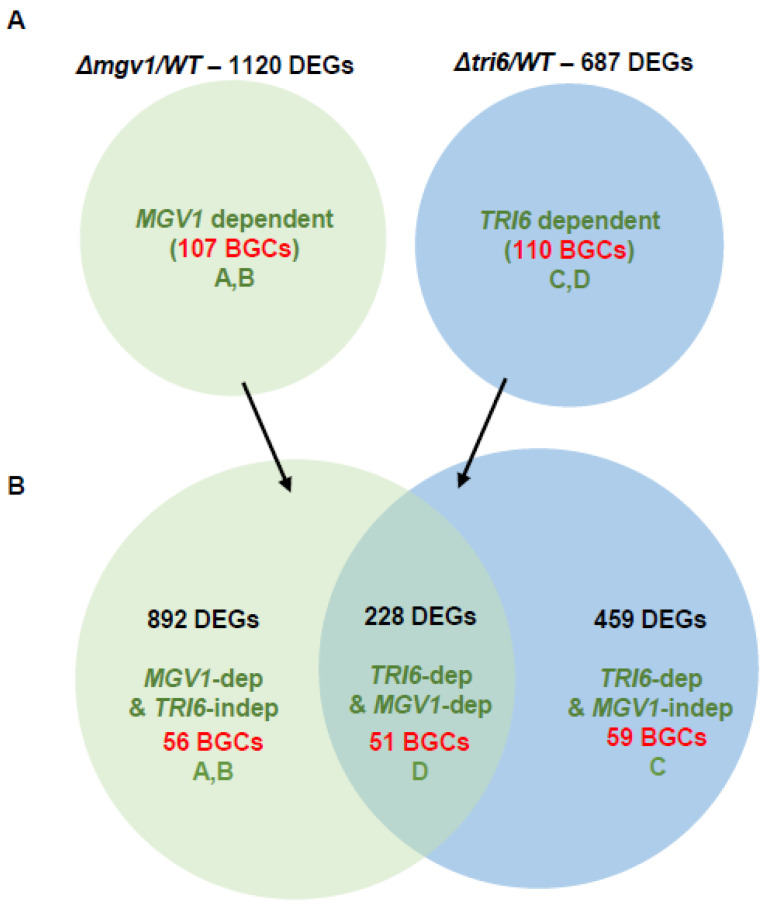
A Venn diagram to distinguish DEGs and BGCs regulated by *TRI6* and *MGV1*. (**A**) datasets from *MGV*-specific regulated genes (Δ*mgv1*/WT; 1120 DEGs) and *TRI6*-specific regulated genes (Δ*tri6*/WT; 687 DEGs) were compared and identified both DEGs and BGCs regulated either independently or dependent of *MGV* and *TRI6*; (**B**) for example, 228 DEGs are co-regulated by both MGV1 and TRI6 also identifies the pathways and corresponds to pathway D. The details on the DEGs and BGCs are presented in Appendix A.

**Figure 7 jof-09-00816-f007:**
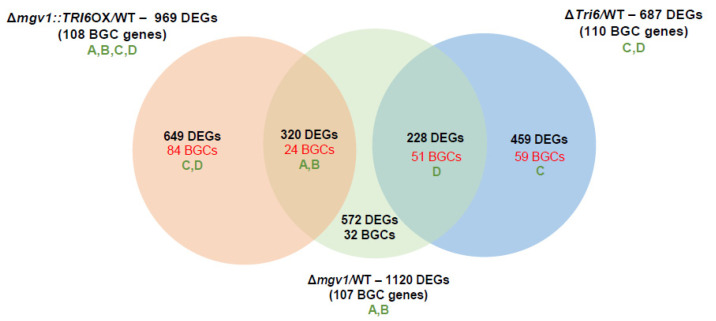
Venn diagram to identify epistatic regulation of BGCs. Epistatic regulation of BGCs between *MGV1* and *TRI6* was established by constitutive overexpression of *TRI6* in the Δ*mgv1* mutant strain (Δ*mgv1*::*TRI6*OX/WT). DEGs and BGCs were compared between the Δ*mgv1*::*TRI6*OX/WT and the Δ*mgv1*/WT strains. A, B, C, and D refers to the signaling pathways regulated by *MGV1* and *TRI6* as outlined in Figure 1.

**Figure 8 jof-09-00816-f008:**
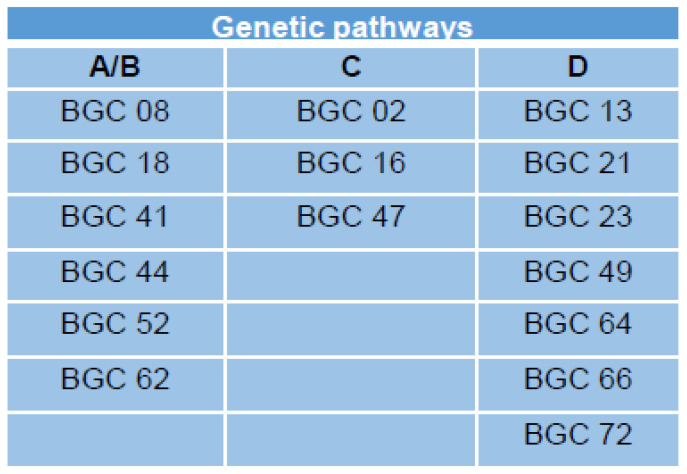
A summary of BGCs regulated by the regulatory hierarchy of *MGV1* and *TRI6* in *F. graminearum*. The BGCs activated through *MGV1* signaling (A/B) by *TRI6* (B) that is either independent of *MGV1* (C) or dependent on *MGV1* (D) are indicated.

## Data Availability

Data is provided in the Appendix A.

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
