# Peer review of "Epistatic Relationship between MGV1 and TRI6 in the Regulation of Biosynthetic Gene Clusters in Fusarium graminearum"

_jof, 2023, doi:10.3390/jof9080816_

Round 1
Reviewer 1 Report
Please see recommendations given below:
In line 98; is it 24 h?
In line 109; please write the name(s) of enzyme(s) used for protoplast isolation? (sigma’s lytic enzyme, novozyme or else or combination?)
Line 110: please provide details for WGS? Fig S3 is not clear, please enhanve the quality of Fig S3
Please provide corrrelation matrix by Pearson or Spearman for qPCRs
Parametric or non parametric data for qPCRs?
Capture for Fig 1; please give F. graminearum and Fusarium in italics.
Please give the names of software sor total package for RNAseq analysis such as trimming, quality control, mapping, aligning, counting etc. (before R data mining). If CLC is complete system for RNAseq analysis such as RASflow or Pigx, please give details for CLC.
Please give names for R packages in addition to DESeq2 such as apelglm, pheatmap, tidyverse, ggrepel etc. For obtaining detailed figures and analysis in R based data mining.
For 2.3 and 2.4.; please give information about DNaseI treatment and no RT PCR.
Fig 4 should be presented with high level of quality
Please provide volcano plot for RNAseq data
Discussion section is not presented in details, please talk about the previous studies including mgv1 and tri6 (potential acceptance of these two genes asmarker genes), also please give more papers published in recent years including the usefulness of RNAseq analysis in F. graminearum such as given below:
https://doi.org/10.1016/j.fgb.2021.103518
https://doi.org/10.1111/ppa.13768
Author Response
In line 98; is it 24 h? – yes, changed accordingly
In line 109; please write the name(s) of enzyme(s) used for protoplast isolation? (sigma’s lytic enzyme, novozyme or else or combination?) – have provided our previously published manuscript as reference.
Line 110: please provide details for WGS? – Lines 115-118. Fig S3 is not clear, please enhance the quality of Fig S3 - Done
Please provide corrrelation matrix by Pearson or Spearman for qPCRs; we use Pfaffl method (added new reference) as mentioned in line 148. This method takes into account correlation matrix by Pearson.
Parametric or non parametric data for qPCRs? – assumes normal distribution - The Pfaffl method
Capture for Fig 1; please give F. graminearum and Fusarium in italics. Done
Please give the names of software sor total package for RNAseq analysis such as trimming, quality control, mapping, aligning, counting etc. (before R data mining). If CLC is complete system for RNAseq analysis such as RASflow or Pigx, please give details for CLC. info is provided in lines 130-135
Please give names for R packages in addition to DESeq2 such as apelglm, pheatmap, tidyverse, ggrepel etc. For obtaining detailed figures and analysis in R based data mining.
For 2.3 and 2.4.; please give information about DNaseI treatment and no RT PCR (Reference 25)
Fig 4 should be presented with high level of quality - Done
Please provide volcano plot for RNAseq data
Volcano plots enable visual identification of genes with significant fold changes in the RNA-seq data, the focus of this manuscript was mainly on the BGCs within the DEGs. Since we have large number of RNA-seq data, we feel that volcano plots for each of them would not be very useful. Moreover, a lengthy discussion of genes that do not belong to the BGCs would dilute the impact of our findings. Nevertheless, we are currently analyzing the datasets in-depth of DEGs with respect to the KEGG pathways. This might shed more light into the pathways linking MGV to primary and other regulatory pathways.
Discussion section is not presented in details, please talk about the previous studies including mgv1 and tri6 (potential acceptance of these two genes asmarker genes), also please give more papers published in recent years including the usefulness of RNAseq analysis in F. graminearum such as given below:
We have made additional observations that ties previous studies (see highlighted in the discussion section)
https://doi.org/10.1016/j.fgb.2021.103518
https://doi.org/10.1111/ppa.13768

Reviewer 2 Report
In this manuscript, the authors distinguish the BCGs regulated by Mgv1 and those regulated by Tri6 through comparative transcriptome analysis. It is of crucial scientific significance. The manuscript is technically sound with potential for publication. My major confusion is the difference between Pathway B and Pathway D. Some figures confuse me. In Figure 2, pathway D seems to be downstream of pathway B. But in the main text and the figure legend, it seems not. In Figure 6, Δmgv1/WT was labeled “A, B, D”, while Δtri6/WT was labeled “B, C”, the same data in Figure 7 was labeled “B, D”. If I did not miss some important information, this should be addressed clearly.
Other comments:
The abbreviation should be unique and consistent in the manuscripts.
Both RT-qPCR (in the main text) and qRT-PCR (in Figure S4) were used to describe the quantitative PCR. The former is preferred.
The term BGCs is abbreviated for biosynthetic gene clusters. However, simultaneously, it stands for genes in the BGC in the main text (especially the RNA-seq analysis part). For example, in line 343, “459 DEGs with 59 BGCs”.
The name of a BCG should be in one form of BCG xx or BCGxx, not all. Like “BCG18” in Line 371, while “BCG 18” in Line 398.
Line 91: the species of which the GAPDH gene is should be given.
The medium recipe should be given in the Materials and Methods section.
Line 97: GYEP medium
Line 174: growth medium
Line 180: second-stage media
Line 98: “as described” means as in the PN condition or others?
Line 112: “(RT-qPCR” lacks a right bracket
Line 119: the full name of “RNP complex” should be given.
Line 139-140: “(RT-qPCR)”, brackets are unnecessary.
Line 152 and Line 160: I am not an expert in statistical analysis. I wonder to know why n=16 or n=24. There are not two or three independent experiments?
Line 335: “BCGs 13” means “BCG 13”?
Line 422 “BCG16s” means “BCG 16”
Figure 5 lacks the significance label.
Author Response
Reviewer 2:
In this manuscript, the authors distinguish the BCGs regulated by Mgv1 and those regulated by Tri6 through comparative transcriptome analysis. It is of crucial scientific significance. The manuscript is technically sound with potential for publication. My major confusion is the difference between Pathway B and Pathway D. Some figures confuse me. In Figure 2, pathway D seems to be downstream of pathway B. But in the main text and the figure legend, it seems not. In Figure 6, Δmgv1/WT was labeled “A, B, D”, while Δtri6/WT was labeled “B, C”, the same data in Figure 7 was labeled “B, D”. If I did not miss some important information, this should be addressed clearly. Apologies for the mixup. The Figure, legend, and the text has been corrected. Line 253; 259
Other comments:
The abbreviation should be unique and consistent in the manuscripts. Done
Both RT-qPCR (in the main text) and qRT-PCR (in Figure S4) were used to describe the quantitative PCR. The former is preferred. Has been corrected
The term BGCs is abbreviated for biosynthetic gene clusters. However, simultaneously, it stands for genes in the BGC in the main text (especially the RNA-seq analysis part). For example, in line 343, “459 DEGs with 59 BGCs”. Has been corrected
The name of a BCG should be in one form of BCG xx or BCGxx, not all. Like “BCG18” in Line 371, while “BCG 18” in Line 398. Has been corrected
Line 91: the species of which the GAPDH gene is should be given. Has been corrected
The medium recipe should be given in the Materials and Methods section. We have modified this part for clarity. lines 95-103.
Line 97: GYEP medium; Line 174: growth medium; Line 180: second-stage media
Line 98: “as described” means as in the PN condition or others? Done
Line 112: “(RT-qPCR” lacks a right bracket Done
Line 119: the full name of “RNP complex” should be given. Done
Line 139-140: “(RT-qPCR)”, brackets are unnecessary. Done
Line 152 and Line 160: I am not an expert in statistical analysis. I wonder to know why n=16 or n=24. There are not two or three independent experiments? As indicated in line 156, two biological replicates were conducted with 16 tech reps per experiment. Similarly, in line, 164, three biol replicates with 24 tech reps in each experiments. A large number of tech reps reduces errors that prone for these type of expts.
Line 335: “BCGs 13” means “BCG 13”? Has been corrected
Line 422 “BCG16s” means “BCG 16” Has been corrected
Figure 5 lacks the significance label. Has been corrected line 315 and in the figure
